# The Influence of the COVID-19 Event on Deviant Workplace Behavior Taking Tianjin, Beijing and Hebei as an Example

**DOI:** 10.3390/ijerph18010059

**Published:** 2020-12-23

**Authors:** Yingyan Liu, Zaisheng Zhang, Heng Zhao

**Affiliations:** College of Management and Economics, Tianjin University, Tianjin 300072, China; zhangzs@tju.edu.cn (Z.Z.); zhaohenglg@126.com (H.Z.)

**Keywords:** novel coronavirus pneumonia, Corona Virus Disease 2019 (COVID-19), event system theory, emotional exhaustion, constructive deviant behavior, destructive deviant behavior

## Abstract

*Background*: Since the beginning of 2020, the Corona Virus Disease has broken out globally. This public health incident has had a great impact on the work and life of the public. *Aim*: Based on the event system theory, this article explored the influence of the “COVID-19” event on emotional exhaustion and deviant workplace behaviors. Methods: This survey’s objects are employees working in Tianjin, Beijing, Hebei affected by the epidemic. Using the questionnaire star, the online platform of the Marketing Research Office of Peking University and “snowball” methods 700 questionnaires were collected. *Results*: The response rate was 89.71% (n = 700). Female employees are more sensitive to the perceived event strength of the novel coronavirus pneumonia than male employees (F = 10.94, *p* <0.001); Employees aged 30–40 affected by the epidemic have the highest level of emotional exhaustion (F = 5.22, *p* < 0.01); A higher education level leads to a higher level of emotional exhaustion (F = 4.74, *p* < 0.01); The emotional exhaustion is polarized with the annual family income (F = 4.099, *p* < 0.01). *Conclusions*: The novelty, disruption, criticality of the Corona Virus Disease event has had a positive impact on the emotional exhaustion of employees in the workplace; Emotional exhaustion plays a partly mediating role between event strength with constructive deviant behaviors, and destructive deviant behaviors. Emotional exhaustion has a positive effect on creative constructive deviant behaviors, challenging constructive deviant behaviors, and interpersonal destructive deviant behaviors. Emotional exhaustion has a negative impact on organizational destructive deviant behaviors, and has no significant impact on interpersonal constructive deviant behaviors.

## 1. Introduction

Since the outbreak of the Corona Virus Disease (COVID-19), the virus has spread rapidly at home and abroad. Since the attributed start of the event, the novel coronavirus pneumonia pandemic has continued to develop since 23 January 2020, and more than 300,000 new cases are added every day all around world. Regarding the space attribute of the event, the infection has spread to 188 countries and regions around the world, causing nearly 1.6 million deaths worldwide. There have been more than 90,000 confirmed cases in China, and all provinces (municipalities and autonomous regions) across the country have been affected, up to December 2020. Since the outbreak, governments all over the world are doing their best to deal with the epidemic, save lives, reduce the impact of the epidemic on the economic development and social governance of their countries. In order to ensure the economic operation of various countries, many countries have launched online office modes. Local public officials in China responded to the call of the Party Central Committee to immediately return to their job positions while the masses are consciously isolated at home. Enterprise employees worked online to reduce the impact of the Corona Virus Disease on family income.

In a survey of 4000 teleworkers on Flex Jobs (a US employment website), 95% said their productivity was the same or higher than before the outbreak of COVID-19. Therefore, this research attempts to explores the emotional exhaustion and workplace deviance of employees of different ages, educational backgrounds, and income levels affected by the Corona Virus Disease event, and provides certain countermeasures and suggestions for enterprises and institutions to deal with major emergencies, understand the psychological needs of employees, and appease employees’ emotions to ensure normal operations of the company. There are many reasons influencing employees’ workplace deviance. Firstly, the existing research has verified that employees’ personality, leadership style, team atmosphere and other factors have a profound impact on employees’ work emotions. However, few studies have focused on the impact of external events on employees’ workplace behavior. Different from large-scale public health events in the past, the novel coronavirus pneumonia event is a major passive event. Based on the strength attribute of the event, the public’s perception of the COVID-19 event directly affects their behavior [1]. Secondly, according to the event system theory, negative events are positively correlated with negative emotions [2]. Negative emotions can cause symptoms such as physical fatigue, insomnia, headaches, etc. [3], which can lead to feelings of helplessness, anxiety, and social decline, job unsatisfaction, breaking organizational norms and other behaviors [4,5]. This major public health event disrupted the previous lifestyle of the public, changed the previous office style of employees, and caused emotional exhaustion. Therefore, this article explores the feelings of employees of different genders, ages, educational backgrounds and income levels about the COVID-19 event, and the mechanism(s) influencing employees’ perceptions of novelty, disruption, criticality of the Corona Virus Disease event with their possible destructive or constructive deviant behaviors, which should help companies understand employees’ psychological needs and practical difficulties, reduce job burnout and stimulate work enthusiasm.

The contributions of this study lie in: firstly, the study of workplace deviant behavior in this paper is based on event system theory, and taking the Corona Virus Disease which has broken out globally as research back-ground, it expands the application scope of event system theory, and enriches the empirical research of the theory. Secondly, taking the external events as the breakthrough point, this article discusses the impact of external major events on employees’ psychology and behavior. Thirdly, this paper attempts to break through the existing research on workplace deviance, and explores the compatibility of two types of deviant behavior. In other words, employees who implement destructive deviant behavior may also display behavior beneficial to the organization, and vice versa.

## 2. Theoretical Basis and Research Hypotheses

### 2.1. Event System Theory

The previous literature has explored the relationship among stable features within entities from the macro and micro perspectives, ignoring the dynamic impact of events on the entity. Event system theory is the study of individual behaviors under the combined effects of event strength, space and time [1]. The novel coronavirus pneumonia event affects people’s work and life based on the three attributes of event time, space, and strength. In terms of time attribute of event, the more it affects the development of the individual and the longer the duration, the greater impact on the individual. In terms of the space attributes of the event, it includes four dimensions: origin, vertical spread range, horizontal spread range, and distance to individuals. The novel coronavirus pneumonia event covers various cities in China, with a large spread and a wide range of spread. Therefore, the time and space attributes of the event are relatively stable, and there is little difference between individuals in the same city, however, the attributes of event strength are different. Individuals have different perceptions of the novelty, disruption and criticality of the event, which greatly affects their behavior in response to the epidemic. Specifically, the novelty of event refers to how the event is different from previous events, and the more the novelty, the more it arouses individuals’ attention, changing their behavior. The criticality of an event refers to the degree of influence on the goals of the enterprise and organization, and the more criticality of an event, the more the individual needs to pay attention to the development process of the incident, and more actively mobilize resources to deal with it. The disruption of event refers to the degree of change and disturbance to the individuals’ past life and habitual coping styles. The higher the disruption, the more individuals need to adjust their existing behavioral patterns [1].

Since the outbreak of the Corona Virus Disease, people take the initiative to isolate themselves at home in China and even around the world, the performance of most companies has been greatly impaired, and the economic situation has declined significantly. Employees’ perception of the strength of novel coronavirus pneumonia event also has a great impact on their mental state and work efficiency, but the impact of different structural dimensions of the strength stimulus on individuals remains to be investigated. Therefore, this study is based on the event system theory to quantitatively analyze employees’ perceptions of the strength of the Corona Virus Disease. This paper intends to explore the impact of event strength on the mental state of employees of different genders, ages, educational backgrounds and income levels, and the mechanism(s) influencing the implementation of deviant workplace behaviors.

### 2.2. The Event Strength Has Positive Effect on Emotional Exhaustion

Emotional exhaustion is the core dimension of job burnout. It is defined as the excessive consumption of emotional resources of employees, leading to depression, haggardness and depression [4]. Research on emotional exhaustion has found that when roles are overloaded and internal resources are insufficient, employees are prone to burnout. In the workplace, male managers have the highest level of emotional exhaustion, followed by female managers. Female employees are more likely to suffer emotional exhaustion than male employees, and young employees are more likely to suffer emotional exhaustion that older ones. Single employees have higher job burnout at work [4]. Long-term emotional exhaustion can cause individual physical exhaustion, insomnia, headaches and other symptoms [6], which can cause feelings of helplessness, despair, anxiety, and separation from the group.

The global economy has been hindered since the outbreak of novel coronavirus pneumonia [7], and during the epidemic, 22.3% of companies reduced their operating pressures by cutting staff and salaries, and 15.8% even halted work completely, which had a major impact on the psychological state of employees [8]. Due to opaque information and excessive negative public opinion, employees have a negative attitude towards personal employment prospects, business management level, and macroeconomic situation [9]. The perceived strength of novel coronavirus pneumonia event has a profound impact on the physical and mental health of employees [10]. According to the event system theory, the event strength includes novelty, disruption, criticality, and strength stimuli of different latitudes affect the individual’s mental state. Therefore, we pose the following hypotheses:

**H1a.** 
*The event novelty has a positive effect on emotional exhaustion.*


**H1b.** 
*The event criticality has a positive effect on emotional exhaustion.*


**H1c.** 
*The event disruption has a positive effect on emotional exhaustion.*


### 2.3. The Mediating Effect of Emotional Exhaustion between Event Strength and Deviant Behavior

Deviant behavior is defined as an employee’s deliberate violation of organizational norms in the workplace based on selfish or altruistic motives. It is a conscious and purposeful subjective behavior that has a negative/positive effect on organizational performance and organizational members. Specifically, constructive deviant behaviors refer to behaviors that employees actively take to violate organizational norms in order to enhance the well-being of the organization or its members [5]. Destructive deviant behavior refers to the behavior of internal employees who deliberately violate organizational norms on other members of the organization. Such behavior will cause damage to the interests of other colleagues in the organization and even the entire collective [11].

Galperin divides constructive deviant behavior into three dimensions: (1) Innovative constructive deviant behavior refers to the behavior of helping the organization in an innovative and non-traditional way, including five items “develop new ways to solve problems” etc. [5]; (2) Challenging constructive deviant behaviors refer to employee behaviors that break or openly challenge established norms in order to help the organization, and includes six items such as “disturbing or breaking the rules in order to complete the work” etc.; (3) Interpersonal constructive deviant behaviors refer to employees’ deviant behaviors against other members of the organization, which includes five items such as “disagreeing with the opinions of others in the working group in order to improve existing work procedures”, etc.. Past literature has found that employees with outgoing personality [12] and higher income levels have a higher perception of fairness, which encourages them to implement constructive deviance [13].

Robinson developed a scale of destructive deviant behaviors with 19 items [14], including seven items on interpersonal deviance such as “being rude to others, forming gangs”, etc. [15], and 12 items on organizational deviance such as “arrive late, leave early, resign”, etc. Studies have shown that employees with violent tendencies, low levels of education, and shorter years of work experience show disregard for organizational rules [16], and reduced compensation can lead to destructive deviant behaviors by employees.

Under the global prevalence of the novel coronavirus pneumonia, the general public has consciously self-quarantined, but employees will inevitably touch unknown people during commuting and work, worrying about their own health, and be unable to balance work and health, causing them to suffer emotional exhaustion. Emotionally exhausted employees find it difficult to provide efficient work performance. Negative work attitudes will cause low work completion, and workers will be late, leave early, complain about leadership, and even resign [17]. When the individual’s emotional resources are exhausted, the employee’s creativity and challenge ability decrease [18]. When employees are highly motivated in their work, they will implement some behaviors in order to improve the existing work procedures, such as making suggestions and negating the opinions of other members of the working group [5]. This shows that employees with high emotional exhaustion are more sensitive to internal and external stimuli in the organization, and the surge in emotional exhaustion induces deviant behaviors in the workplace. Therefore, we pose the following hypotheses:

**H2a.** 
*Emotional exhaustion has a mediating effect between event strength and constructive deviant behavior.*


**H2b.** 
*Emotional exhaustion has a mediating effect between event strength and destructive deviant behavior.*


### 2.4. The Emotional Exhaustion Has Positive Effect on Deviant Behavior

Positive emotions of employees are significantly related to constructive deviant behaviors in the workplace [19]. Employees actively breaking the previous rules to improve work efficiency is a manifestation of organizational health [20], When employees’ happiness is higher, their creativity is stronger, their innovation performance is higher and they are less likely to resign. Constructive deviant behaviors of employees in the workplace are crucial to the survival and development of the organization [21]. In the context of the novel coronavirus pneumonia, employees have suffered a huge psychological impact. Negative emotions such as depression and anxiety have reduced their work input and production efficiency, which has had a negative impact on the development of companies [22]. When employees are in a state of emotional exhaustion, it is difficult to ensure that their behavior can meet organizational expectations and system requirements. The accumulation of negative emotions, such as depression, decadence, fatigue, fear, and tension, will lead to decreased work motivation [23], or even absenteeism without reason, passive laziness, complaining, shirking responsibility and other behaviors [24]. Therefore, we propose that:

**H3a.** 
*Emotional exhaustion has a negative effect on innovative constructive deviance.*


**H3b.** 
*Emotional exhaustion has a negative effect on challenging constructive deviance.*


**H3c.** 
*Emotional exhaustion has a negative effect on interpersonal constructive deviance.*


**H3d.** 
*Emotional exhaustion has a positive effect on organizational destructive deviance.*


**H3e.** 
*Emotional exhaustion has a positive effect on interpersonal destructive deviance.*


Based on the above analysis, this paper constructs a conceptual model of the influence mechanism of event strength on deviant behaviors, as shown in Figure 1:

## 3. Research Design

### 3.1. Research Sample

Affected by the epidemic, this study used online questionnaires to collect and obtain data. The survey objects are employees working in the company. The survey content includes event strength, emotional exhaustion, constructive deviant behavior, destructive deviant behavior, and basic personal information, including gender, age, education, income, industry, etc.

In order to ensure the validity, authenticity and reliability of the information obtained in the research, this research has adopted a number of control measures to strictly control all links in the research process. First of all, the survey participants were informed about the academic purpose of the survey in the initial guidance of the questionnaire, and promised that all materials will be used only for academic research, and the content of the answers will be strictly anonymous and confidential, thereby eliminating the concerns of the survey participants; Secondly, this survey used the questionnaire star and the plat-form of the Marketing Research Office of Peking University to collect data, and adopted the “snowball” method to collect questionnaires. “Snowball” means that the researchers contacted the staff of institutions, state-owned enterprises, and private enterprises in the Tianjin-Beijing-Hebei region, asking them to fill in the questionnaire, and then send it to their friends or other colleagues in their organization to participate in the survey; Finally, setting the answering time, controlling each item to be no less than 3 s, and counting the time it takes to answer the entire questionnaire, and eliminate the questionnaires that are not filled in carefully. A total of 700 questionnaires were returned in this survey. After excluding invalid questionnaires due to factors such as too short answering times, in-complete filling, and continuous answering with the same number, 628 valid questionnaires were obtained, for an effective response rate of 89.71%. The descriptive statistical in-formation is shown in Table 1.

### 3.2. Measuring Tools

The questionnaire design part of this study mainly includes five aspects: the event strength scale designed by Morgeson [1]; the emotional exhaustion scale designed by Maslach [4]; the constructive deviant behavior scale designed by Galperin [5]; the destructive deviant behavior scale designed by Robinson and Bennett [14], shown in Table 2, and the basic demographic information of the respondents. Excluding the basic information, all the questionnaires in this study use Likert’s 7-point method. Interviewed employees need to score 1–7 on all the question items in the event strength scale and emotional exhaustion scale, 1 = “completely disagree”, 4 = “neutral”, 7 = “completely agree”. In the destructive deviant behavior scale and the constructive deviant behavior scale, the respondent was asked to score 1–7, 1 = “completely inconsistent”, 4 = “fair”, 7 = “completely consistent”.

## 4. Result Analysis

### 4.1. Homologous Deviation Test

In order to avoid the common method deviation from affecting the research results, SPSS 22.0 was used to perform Harman’s Single factor test. It was found that the variance explanation rate of the first factor separated out without rotation was 42.546%, which did not reach 50%. Therefore, the sample did not see common method deviation.

### 4.2. Reliability and Validity Test

The reliability and validity of the questionnaire were tested, and SPSS 17.0 was used to calculate the Cronbach’s α of each scale to measure the reliability of the scale. The results found that the Cronbach’s α of event strength scale, emotional exhaustion scale, constructive deviant behavior scale, and destructive deviant behavior scale were all above 0.8, which met the reliability standard, indicating that the questionnaire had good internal consistency. Through the Bartlett test, KMO > 0.8, and *p* < 0.01, indicating that the questionnaire has good structural validity.

### 4.3. Factor Analysis

The maximum variation method was used to rotate the factor load test, select the factors with characteristic root > 1, and perform exploratory factor analysis on the event strength scale, emotional exhaustion scale, constructive deviant behavior scale, and destructive deviant behavior scale. The event strength scale extracts three public factors, the emotional exhaustion scale extracts one public factor, the constructive deviant behavior scale extracts three public factors, and the destructive deviant behavior scale extracts two public factors, and the total interpretation degree of each scale extraction factor is much higher than 50%. Therefore, it is judged that the factors selected by each scale are representative and can explain the overall variables well.

In order to further test the models derived from exploratory factor analysis, this study further confirms factor analysis to compare the fit of competing models. AMOS 17.0 was used to test the discriminative validity between the factors of the model, and compare the nine factors (event novelty, event disruption, event criticality, emotional exhaustion, innovative constructive deviance, challenging constructive deviance, interpersonal constructive deviance, organizational destructive deviance, interpersonal destructive deviance) conducted confirmatory factor analysis and found that the fitting index of the nine-factor model (χ2 = 898.42, df = 292, TLI = 0.967, CFI = 0.957, RMSEA = 0.027) was significantly better than other competitive models. Each factor CR was >0.7 and AVE > 0.5, indicating that the questionnaire has good convergence validity.

### 4.4. Correlation Analysis

In order to avoid the collinearity problem of variables, the correlation coefficient between variables is tested first, and the mean and standard deviation of event strength, emotional exhaustion, constructive deviant behavior, and destructive deviant behavior are calculated. To judge the correlation between the variables, the correlation coefficient ∣r∣ tends to 1, the more relevant, the closer to 0, the less relevant. See Table 3 for details.

It can be seen from Table 3 that there is no collinearity problem among the variables, so the following structural equation model test can be carried out to further explore the relationship between the variables.

### 4.5. Analysis on the Difference of Demographic Variables

The results of the differences of demographic variables showed that, there are significant differences in perception of event strength between female and male employees (F = 10.94, *p* < 0.001). Among the respondents, there were 339 female employees and 289 male employees, and female employees (M = 6.59, SD =0.62) were more sensitive to the perceived event strength of the novel coronavirus pneumonia than male employees (M = 6.27, SD = 0.77); There are significant difference in emotional exhaustion among employees of different age. Among the respondents, 254 employees were under 30 years old, 138 employees were 30–40 years old, 96 employees were 40–50 years old, and 140 employees were over 50 years old. Employees of different ages have different degrees of emotional exhaustion in the face of the epidemic (F = 5.22, *p* < 0.01), employees aged 30–40 (M = 5.92, SD = 1.17), aged under 30 (M = 5.73, SD = 1.08), aged 40–50 (M = 5.52, SD = 1.14), aged over 50 (M = 5.31, SD = 1.03), Therefore, employees aged 30–40 were most affected by the epidemic events, there are significant differences in emotional exhaustion among employees with different education levels (F = 4.74, *p* < 0.01). The interviewees had 161 college, 312 bachelors, 95 masters, and 60 doctoral degrees. Employees with a master’s degree or above are more affected by the novel coronavirus pneumonia than employees with a bachelor’s degree or below There are differences in emotional exhaustion among employees with different family income (F = 4.099, *p* < 0.01). Among the respondents, employees with monthly incomes of more than 10,000 yuan (M = 4.33, SD = 1.01) emotionally fluctuated greatly due to the epidemic, followed by employees with monthly incomes of 1–2 K (M = 3.94, SD = 0.99). Affected by the epidemic situation, the employees with the lowest emotional fluctuation were those with monthly incomes of 3–5 K (M = 3.35, SD = 0.87) and 6–10 K (M = 3.24, SD = 0.92). Other demographic variables were not significant.

### 4.6. Hypothesis Testing

#### 4.6.1. Testing the Effect of Event Strength on Emotional Exhaustion

From Table 4 and Figure 2, it can be seen that the standardized path coefficient of event novelty on emotional exhaustion is 0.524, *p* < 0.001, which has a significant positive effect, and H1a is valid. The standardized path coefficient of event criticality on emotional exhaustion is 0.574, *p* < 0.001, which has a significant positive effect, so H1b is valid; The standardized path coefficient of event disruption on emotional exhaustion is 0.593, *p* < 0.001, which has a significant positive effect, so H1c is valid.

#### 4.6.2. Testing the Mediating Effect

It can be seen from Table 5 that the standardized path coefficient of event strength to emotional exhaustion is 0.624, *p* < 0.001; The standardized path coefficient of emotional exhaustion to constructive deviant behavior is 0.205, *p* = 0.001; The standardized path coefficient of emotional exhaustion to destructive deviant behavior is 0.139, *p* = 0.019; The normalized path coefficient of event strength to constructive deviant behavior is 0.435, *p* < 0.001; The normalized path coefficient of event strength to destructive deviant behavior is 0.512, *p* < 0.001.

According to the mediation test method, we should first analyze the influence of independent variable event strength on dependent variable deviant behavior. If the relationship between the dependent variable: constructive deviant behavior, destructive deviant behavior and the independent variable: event strength is not significant, we stop testing the mediating effect. Secondly, explore whether the independent variable (event strength) affects the mediating variable (emotional exhaustion). If this relationship is not significant, stop testing the mediating effect. Finally, whether both the independent variable and the intermediate variable have an effect on the dependent variable is tested. If the event strength has no significant effect on deviant behaviors, and emotional exhaustion has a significant effect on constructive deviant behaviors and destructive deviant behaviors, it is judged as a complete mediating effect. If the independent variable has a significant effect on the dependent variable, it is judged to be a partial mediator.

In this study, AMOS was used to test the BOOTSTRAP mediation effect. The analysis results are shown in Table 6. The total impact of event strength on constructive deviant behavior is 0.571, the range of bias-corrected is 0.471–0.655, and the range of percentile is 0.47–0.655, both excluding 0, it indicates that the event strength has a significant overall positive effect on constructive deviant behavior; The total impact of event strength on destructive deviant behavior is 0.521, the range of bias-corrected is 0.408–0.627, and the range of percentile is 0.408–0.627, all excluding 0, indicating that event strength has a significant positive effect on destructive deviant behavior. The first step of the mediating test passed.

By examining the indirect effect of event strength through emotional exhaustion on constructive deviant behavior, the indirect effect is 0.059, the range of bias-corrected is 0.006–0.125, and the range of percentile is 0.003–0.122, all excluding 0, which shows that the event strength has a significant indirect positive effect on constructive deviance through the mediating variable emotional exhaustion; In addition, the indirect effect of event strength on destructive deviance through emotional exhaustion is 0.087, the range of bias-corrected is 0.018–0.168, and the range of percentile is 0.016–0.167, all excluding 0, indicating that event strength has a significant indirect positive effect on destructive deviance through emotional exhaustion. Thus, the second step of the mediating test passed.

Finally, examine the direct effect of event strength on constructive deviance. The direct effect is 0.512, the range of bias-corrected is 0.397–0.609, and the range of percentile is 0.402–0.611, all without 0, indicating the event strength has a significant direct positive effect on constructive deviance, so it shows that emotional exhaustion has a partially mediating role between event strength with constructive deviance, so H2a is valid; The direct effect of event strength on destructive deviance is 0.435, the range of bias-corrected is 0.31–0.542, and the range of percentile is 0.31–0.541, all excluding 0, indicating that event strength has a significant direct positive effect on destructive deviance. Therefore, it shows that emotional exhaustion has a partial mediating role between event strength with destructive deviance. H2b is thus valid.

#### 4.6.3. Testing the Effect of Emotional Exhaustion on Deviant Behavior

It can be seen from Table 7 and Figure 2, the standardized path coefficient of emotional exhaustion on innovative constructive deviance is 0.408, *p* < 0.001, which has a significant positive effect, therefore H3a is not valid. The standardized path coefficient of emotional exhaustion on challenging constructive deviance is 0.672, *p* < 0.001, which has a significant positive effect, so H3b is not valid. The standardized path coefficient of emotional exhaustion on interpersonal constructive deviance is 0.232, *p* = 0.015, which has no significant effect, so H3c is not valid. The standardized path coefficient of emotional exhaustion on organizational destructive deviance is −0.711, *p* < 0.001, which has a significant negative effect, so H3d is not valid. The standardized path coefficient of emotional exhaustion on interpersonal destructive deviance is 0.482, *p* < 0.001, which has a significant positive effect, so H3e is valid.

## 5. Conclusions and Discussions

### 5.1. Research Conclusions

This research is based on the event system theory, starting from the event strength perceived by the public, and exploring the changes in employees’ psychological state and their workplace behaviors after being stimulated by the external environment. In this study, when China’s novel coronavirus pneumonia was gradually brought under control and enterprises gradually resumed work and production, 628 valid questionnaires from employees of enterprises in the Tianjin-Beijing-Hebei region were collected online, and our empirical research found that: (1) Female employees are more sensitive to the perceived event strength of the novel coronavirus pneumonia than male employees; (2) Young and middle-aged employees aged 30–40 have the highest level of emotional exhaustion and the greatest pressure, and employees over 50 have the least emotional fluctuations affected by the epidemic; (3) The higher the education level, the higher the perception of event strength, leading to higher levels of emotional exhaustion; (4) The emotional exhaustion of employees in the workplace is polarized by the annual family income; (5) The event novelty, event disruption, event criticality of the novel coronavirus pneumonia has a positive effect on employees’ emotional exhaustion; (6) Emotional exhaustion plays a partially mediating role between event strength with constructive deviance and destructive deviance; (7) Although employees affected by the epidemic are immersed in emotional exhaustion, their innovative constructive deviance and challenging constructive deviance have generally increased. Employees actively or passively must break the original working methods, adapt to the external environment, and improve work efficiency. Emotional exhaustion of employees has no significant impact on interpersonal constructive deviant behaviors. Due to the limitation of working space, employees cannot interact in time. The interviewed employees mostly adopt methods such as reducing their own role conflicts, dividing work-family boundaries, adapting to new working methods, improving economic security, and even challenging new fields of work to deal with the impact of the novel coronavirus pneumonia on family life, and balance their anxiety [25]; Emotional exhaustion has a significant positive effect on interpersonal destructive deviance, but has a negative effect on organizational destructive deviance. Affected by the epidemic, global economic development has stagnated, and a large number of companies have implemented large-scale layoffs in order to reduce operating costs, resulting in a surging unemployment rate. Freelancers were forced to close their shops and suffered heavy economic losses. In the face of this major global public health incident, under the severe economic situation, young and middle-aged employees are burdened with financial pressures such as mortgages, car loans, and family burdens. Even if employees are nervous, fearful and anxious, they will not implement organizational deviant behaviors such as being late, leaving early, or leaving. On the contrary, most of the interviewed employees will work harder, hoping not to be laid off by the company, and resolve their inner dissatisfaction through interpersonal deviance tools such as complaints and cliques [26].

### 5.2. Discussions

This study found that under the stimulation of major events outside the organization, female employees, young and middle-aged employees, highly educated employees, and employees with higher or lower income levels were more sensitive to the epidemic situation, and emotional exhaustion positively affected interpersonal destructive deviance. This is consistent with the findings of theoretical research on emotional events [27]. The emotional event theory believes that employees will inevitably encounter events that prompt them to produce positive or negative emotions at work. Due to emotional mobilization, the behavior of employees will change, but different individuals have different emotional responses to the same event. In the comparative study of job burnout between management and non-management, it is found that emotional exhaustion of management employees is higher than that of non-management employees, and the level of emotional exhaustion of female employees is higher than that of male employees [28]. Workplace experience will affect employees’ motivation, behavior, and work performance through the medium of emotions. When employees encounter setbacks, they will produce negative emotions and show more destructive deviant behaviors and aggressive behaviors at work [29]. When employees lack organizational support, managers do nothing, and colleagues shirk their responsibilities, they will suffer emotional exhaustion, and take actions to repair themselves [30], venting and retaliating against organizational property, organizational environment, and organizational members [31], through manifestations such as stealing, destroying public property, insulting colleagues, sabotage, arriving late and leaving early, resigning and other destructive deviant behaviors [32], but positive emotions can trigger more creative thinking and behaviors [33], making individuals more focused and flexible in problem-solving, more willing to communicate and collaborate with other members of the organization, and improve work efficiency [34]. Positive emotions can motivate employees to break the stereotypes, dare to challenge, and better adapt to the external environment [35].

The contribution of this article lies in the exploration of the unconventional changes in employee emotions and workplace behaviors under the epidemic situation based on the event system theory. Normally, when the employees in the organization have negative emotions, their innovation consciousness gradually declines, and most employees tend to stick to the stereotypes. From previous studies, it is also difficult to judge that major events outside the organization can prompt employees to actively innovate and challenge themselves. However, the current research results reveal that in the context of the outbreak of the global novel coronavirus pneumonia, employees have internalized emotional exhaustion into work motivation, promoted innovative constructive deviance, challenging constructive deviance, cherished job opportunities, and reduced destructive organizational deviance (such as: over-reimbursement, lateness, absence, resignation, etc.). This article supplements and enriches the relevant research on workplace deviant behaviors, and provides suggestions for companies to reasonably ease employee emotions and balance labor costs.

### 5.3. Management Implications

Some actions companies can take to address this situation are:

#### 5.3.1. Establish “Employee Care Plan” to Relieve Employees’ Negative Emotions

In the face of a sudden epidemic situation, enterprises should popularize epidemic prevention knowledge, provide epidemic prevention supplies for employees who come to work during the epidemic period, and implement isolated office work. Enterprises should also pay special attention to female employees, young and middle-aged employees, highly educated employees, high-income groups and social bottom groups, so as to control the level of human capital in enterprises. According to the physical condition of the women in lactation period, the home office mode should be adopted to complete the task. Young and middle-aged employees shoulder the economic pressure of caring for the elderly, raising children, and even housing and car loans. They are also the backbone of the enterprise. Young and middle-aged employees should be encouraged to turn the pressure into motivation. According to the company’s own situation, give transportation subsidies, meal subsidies, distribution of daily necessities, food and other welfare, show the care of the enterprise for staffs. Cultivate the senior managers of enterprises to establish the awareness of “community of common destiny” and the overall situation of the country, the nation and the enterprise [36]. Try the mode of “sharing employees” in cooperation among enterprises [37], reduce the labor cost of enterprises, provide economic security for low-income people, and reduce the turnover rate.

#### 5.3.2. Open a Long-Term Communication Channel to Give Employees the Space of Independent Decision-Making

Using Ding Talk, WeChat groups and enterprises’ internal office platform, realize the cooperation and information sharing among departments. Divide the work tasks, form working groups, specify work nodes, submit daily work progress, and establish flat management mode. Pay attention to results, relax the work process, set necessary restrictions, time schedule, and strategic deployment, give employees flexible space to work, and encourage employees to divide work family boundaries [38]. Break the bureaucratic atmosphere in the workplace and realize vertical management. Accept the creativity of employees, recognize the initiative changes made by employees to better complete their work. A special fund should be set up to give monetary or material rewards to employees who have made outstanding contributions to the organization.

### 5.4. Research Limitations

Although this article reveals the influence mechanism of employees’ perception of the external events out of organization on their workplace behavior, it still has certain limitations. First of all, our research uses a questionnaire survey. We have proposed that the relationship between event strength, emotional exhaustion, and deviant behavior, the causal relationship may be reversed. Longitudinal data can be used in the future to eliminate the problem of false regression of hypothetical relationships.

Secondly, the model proposed in this study is not comprehensive. From the related literature, we find that emotional exhaustion is not the only factor that affects the event strength on workplace deviant behavior. There are other factors that can be explored, such as corporate culture, leadership style and employee competence, etc.

Finally, the data we collected are only based on the Tianjin-Beijing-Hebei region. Therefore, some of the findings of this study may have certain differences due to different regions and different strength of the epidemic, which may limit the universality of our research results. Therefore, the sample scope should be expanded in the future to further test the cross-regional differences.

## Figures and Tables

**Figure 1 ijerph-18-00059-f001:**
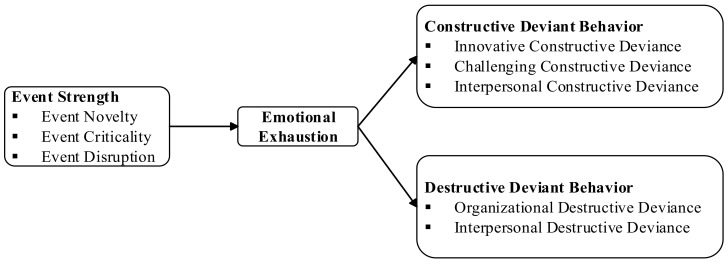
Theoretical model.

**Figure 2 ijerph-18-00059-f002:**
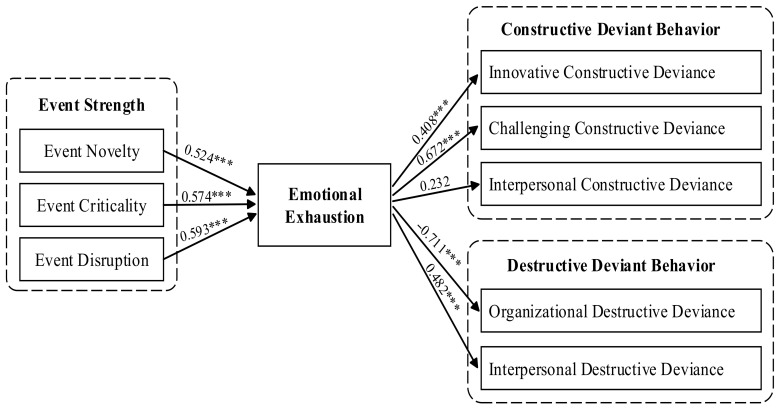
Path test results of the theoretical model; *** *p* < 0.001.

**Table 1 ijerph-18-00059-t001:** Descriptive statistical analysis.

Items	Options	Quantity	Percentage	Items	Options	Quantity	Percentage
Gender	Male	289	46.1%	Income	1–2 K	56	8.94%
	Female	339	53.9%		3–5 K	319	50.84%
Age	<30	254	40.50%		6–10 K	212	33.80%
	30–40	138	21.79%		>10 K	41	6.42%
	40–50	96	15.36%	Industry	Student	63	10.06%
	>50	140	22.35%		Teacher	82	13.13%
Education	College	161	25.70%		Civil servant	96	15.36%
	Bachelor	312	49.72%		Institutions	109	17.32%
	Master	95	15.08%		Enterprise	253	40.22%
	PhD	60	9.50%		Freelance	25	3.91%

**Table 2 ijerph-18-00059-t002:** Scale items.

Scales	Dimensions	Items
Event Strength	Event Novelty	There is a clear, known way to respond to this event (R)
		There is an understandable sequence of steps that can be followed in responding to this event (R)
		Can rely on established procedures and practices in responding to the events (R)
		Had rules, procedures, or guidelines to follow when this event occurred (R)
	Event Criticality	This event is critical for the long-term success of my company
		Dealing with emergencies is the primary event of my company
		Dealing with emergencies is an important event of my company
	Event Disruption	This emergency destroyed the original work capacity (performance) of my company, making the work unable to be completed.
		This emergency made our company stop to think about how to deal with it.
		This emergency has changed our company’s usual response to emergencies.
		The occurrence of this emergency needs our company to change the previous working mode.
Emotional Exhaustion	Emotional Exhaustion	I feel emotionally drained from my work
		I feel used up at the end of the workday
		Working with people all day is really a strain for me
		I feel burned out from my work
		I feel I’m working too hard on my job
		I feel like I’m at the end of my rope
Destructive Deviance	Interpersonal Destructive Deviance	Made fun of someone at work
		Said something hurtful to someone at work
		Made an ethnic, religious, or racial remark at work
		Cursed at someone at work
		Played a mean prank on someone at work
		Acted rudely toward someone at work
		Publicly embarrassed someone at work
	Organizational Destructive Deviance	Taken property from work without permission
		Spent too much time fantasizing or daydreaming instead of working
		Falsified a receipt to get reimbursed for more money than you spent on business expenses
		Taken an additional or longer break than is acceptable at your workplace
		Come in late to work without permission
		Littered your work environment
		Neglected to follow your boss’s instructions
		Intentionally worked slower than you could have worked
		Discussed confidential company information with an unauthorized person
		Used an illegal drug or consumed alcohol on the job
		Put little effort into your work
		Dragged out work in order to get overtime
Constructive Deviance	Innovative Constructive Deviance	Developed creative solutions to problems
		Searched for innovative ways to perform day to day procedures
		Decided on unconventional ways to achieve work goals
		Departed from the accepted tradition to solve problems
		Introduced a change to improve the performance of your work group.
	Challenging Constructive Deviance	Sought to bend or break the rules in order to perform your job
		Violated company procedures in order to solve a problem
		Departed from organizational procedures to solve a customer’s problem
		Bent a rule to satisfy a customer’s needs
		Departed from dysfunctional organizational policies or procedures to solve a problem
		Departed from organizational requirements in order to increase the quality of services or products
	Interpersonal Constructive Deviance	Reported a wrong-doing to co-workers to bring about a positive organizational change
		Did not follow the orders of your supervisor in order to improve work procedures
		Disagreed with others in your work group in order to improve the current work procedures
		Disobeyed your supervisor’s instructions to perform more efficiently
		Reported a wrong-doing to another person in your company to bring about a positive organizational change

**Table 3 ijerph-18-00059-t003:** Variable correlation analysis.

Variables	1	2	3	4	5	6	7	8
1. Gender	1							
2. Age	0.02	1						
3. Education	0.03	0.02	1					
4. Income	0.05	0.48 *	0.721 **	1				
5. Event strength	0.55 *	−0.614 **	0.788 **	0.488 *	1			
6. Emotional exhaustion	0.587 **	−0.738 **	0.756 **	0.516 *	0.424 **	1		
7. Construction deviance	−0.567 *	−0.776 **	0.621 **	0.639 **	0.612 **	0.731 **	1	
8. Destructive deviance	0.631 *	−0.816 **	−0.796 ***	−0.677 **	0.517 **	0.432 *	0.06	1

Note: * stands for *p* < 0.05; ** stands for *p* < 0.01; *** stands for *p* < 0.001.

**Table 4 ijerph-18-00059-t004:** Path test of event strength to emotional exhaustion.

Dependent Variable	Path	Independent Variable	Estimate	S.E.	C.R.	*p*
Emotional exhaustion	<---	Event novelty	0.524	0.047	11.149	***
Emotional exhaustion	<---	Event criticality	0.574	0.051	11.255	***
Emotional exhaustion	<---	Event disruption	0.593	0.059	10.051	***

Note: S.E. (Standard Error); C.R. (Critical Ratio); *p* (*p*-Value); *** stands for *p* < 0.001.

**Table 5 ijerph-18-00059-t005:** Mediating effect test.

Dependent Variable.	Path	Independent Variable	Estimate	S.E.	C.R.	*p*
Emotional exhaustion	<---	Event strength	0.624	0.054	11.556	***
Constructive deviance	<---	Emotional exhaustion	0.205	0.055	3.727	0.001
Destructive deviance	<---	Emotional exhaustion	0.139	0.068	2.044	0.019
Constructive deviance	<---	Event strength	0.435	0.049	8.878	***
Destructive deviance	<---	Event strength	0.512	0.062	8.258	***

Note: S.E. (Standard Error); C.R. (Critical Ratio); *p* (*p*-Value); *** stands for *p* < 0.001.

**Table 6 ijerph-18-00059-t006:** Sectional test of mediation effect.

Dependent Variable	Total Effect	Lower Bounds	Upper Bounds	Lower Bounds	Upper Bounds
Constructive deviance	0.571	0.471	0.655	0.470	0.655
Destructive deviance	0.521	0.408	0.627	0.408	0.627
**Dependent Variable**	**Indirect Effect**	**Lower Bounds**	**Upper Bounds**	**Lower Bounds**	**Upper Bounds**
Constructive deviance	0.059	0.006	0.125	0.003	0.122
Destructive deviance	0.087	0.018	0.168	0.016	0.167
**Dependent Variable**	**Direct Effect**	**Lower Bounds**	**Upper Bounds**	**Lower Bounds**	**Upper Bounds**
Constructive deviance	0.512	0.397	0.609	0.402	0.611
Destructive deviance	0.435	0.310	0.542	0.310	0.541

**Table 7 ijerph-18-00059-t007:** Path test of emotional exhaustion to deviant behavior.

Dependent Variable	Path	Independent Variable	Estimate	S.E.	C.R.	*p*
Innovative constructive deviance	<---	Emotional exhaustion	0.408	0.057	7.158	***
Challenging constructive deviance	<---	Emotional exhaustion	0.672	0.072	9.333	***
Interpersonal constructive deviance	<---	Emotional exhaustion	0.232	0.064	3.625	0.015
Organizational destructive deviance	<---	Emotional exhaustion	−0.711	0.053	−13.415	***
Interpersonal destructive deviance	<---	Emotional exhaustion	0.482	0.052	9.269	***

Note: S.E. (Standard Error); C.R. (Critical Ratio); *p* (*p*-Value); *** stands for *p* < 0.001.

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
