# Peer review of "The Influence of the COVID-19 Event on Deviant Workplace Behavior Taking Tianjin, Beijing and Hebei as an Example"

_ijerph, 2020, doi:10.3390/ijerph18010059_

Round 1
Reviewer 1 Report
In this paper the authors analyzed the impact of COVID-19 on workplace employee emotions and deviant behaviors.
The form, styl and structure of the manuscript is to accept.
English is also accepted, however some moderate changes are required.
I did find any significant drawbacks or flaws.
The topic is general interesting, but I am afraid it brings no significant new knowledge.
Some issues should be raised:
- if it was a prospective study on humans should it be any Bioethics Committee approval? ( no matter that safe questionnaires were used)
- abstract and references do not meet criteria for JERPH- please read instruction for authors and correct it accordingly
- tables must be improved; In tables 2-4 and 8 authors use eg. *,** S.E., C.R.- You have to explain it,
- tables 5-7 should be joined to one table.
Author Response
Dear Reviewer,
On behalf of my co-authors, we thank you very much for giving us an opportunity to
revise our manuscript, and we also appreciate you very much for your
positive and constructive comments and suggestions on our manuscript.
We have studied your comments carefully, according to the requirements of the reviewers, the questions were answered one by one, and the article was carefully revised. All the revisions in the article were highlighted. Because of your suggestions, the revised manuscript will become better and readers will get more valuable information.
Attached please find the revised version and a point-by-point response, which we would like to submit for your kind consideration("Please see the attachment").
Appreciated.
Last, we would like to express our sincere gratitude for your patient help and beneficial suggestions on our paper. Hope fully these revisions could fulfill your requirements.
Yours sincerely
Yingyan Liu

Reviewer 2 Report
Thank you for the opportunity to review this manuscript submission.
I found the study to be very interesting and the methods strong, and i think The topic is very interesting and is needed
However, the manuscript need to do some major revisions.
In Introductuion, I recommend re-writing this brief section to improve the readability and narrative flow.
What is going on in the world? I only find information from China
In ethical consideratoions, you have the answer of some Ethics Committee?
In the discussion it is recommended to use scientific literature more current, because there are many articles published by what we encourage them to perform a search of articles.
There is no contrast with other studies. It does not provide new bibliography.
Author Response

(The authors gave the same response as above.)

Reviewer 3 Report
Lines 127-129 missing references.
Results must be presented for all conclusions, no clear data for conclusions 1-4 can be found.
Scales items are not presented in methodological part (some are presented in theoretical part, some (for ex., event strength) is missing. Sample items presentation could improve understanding of these constructs.
Final model could be presented as a picture of theoretical model with paths.
Is the same model suitable for men and women?
Discussion section - no discussion can be found, just conclusions.
Management implications should be revised with attention to this research results because some of them are not related with results directly and are far away from them.
Author Response

(The authors gave the same response as above.)
